# The role of community engagement in promoting research participants' understanding of pharmacogenomic research results: Perspectives of stakeholders involved in HIV/AIDS research and treatment

**Sylvia Nabukenya**[1,3]* , **David Kyaddondo**[2] , **Ian Guyton Munabi**[1] , **Catriona Waitt**[3,4] , **Adelline Twimukye**[3] , **Erisa S. Mwaka**[1]

**1** Department of Anatomy, College of Health Sciences, Makerere University, Kampala, Uganda, **2** Child Health and Development Centre, College of Health Sciences, Makerere University Kampala, Uganda, **3** Infectious Diseases Institute, College of Health Sciences, Makerere University, Kampala, Uganda, **4** Department of Pharmacology and Therapeutics, University of Liverpool, Liverpool, United Kingdom

☯ These authors contributed equally to this work.
* nabukenyas89@gmail.com

## Abstract

Community engagement (CE) is praised to be a powerful vehicle in empowering communities with knowledge and skills to make informed decisions for better health care. Several CE approaches have been proposed to improve participants' and research communities' understanding of genomic research including pharmacogenomic information and results. However, there is limited literature on how these approaches can be used to communicate findings of pharmacogenomic research to communities of people living with HIV. This study explored stakeholders' perspectives on the role of community engagement in promoting understanding of pharmacogenomic research results among people living with HIV. We adopted a qualitative approach that involved 54 stakeholders between September 2021 and February 2022. We held five focus group discussions among 30 community representatives from five research institutions, 12 key informant interviews among researchers, and 12 in-depth interviews among ethics committee members. A thematic approach was used to analyze the results. Five themes merged from this data and these included (i) benefits of engaging communities prior to returning individual pharmacogenomic research results to participants. (ii) Obtaining community consensus on the kinds of pharmacogenomic results to be returned. (iii) Opinions on how pharmacogenomic research information and results should be communicated at community and individual levels. (iv) Perceived roles of community stakeholders in promoting participants' understanding and utilization of pharmacogenomic research results. (v) Perceived challenges of engaging communities when returning individual results to research participants. Stakeholders opined that CE facilitates co-learning between researchers and research communities. Researchers can adapt existing CE approaches that are culturally acceptable for meaningful engagement with minimal ethical and social risks when communicating pharmacogenomic research results. CE approaches

**Data Availability Statement:** All relevant data are within the manuscript and its Supporting information files.

**Funding:** This study was funded by the Fogarty International Center of the National Institute of Health through the Makerere University International Bioethics Research Training Program Grant Number D43TW010892. The funders had no role in study design, data collection and analysis, decision to publish, or preparation of the manuscript.

**Competing interests:** The authors have declared that no competing interests exist.

can facilitate understanding of pharmacogenomic research and findings among research participants and communities. Therefore, if creatively adapted, existing and new CE approaches can enable researchers to communicate simple and understandable results of pharmacogenomic research.

## Introduction

There is an increase in pharmacogenomic research in sub-Saharan Africa aimed at improving HIV treatment [1–5]. In Uganda, there are about 1.5 million people living with HIV (PLHIV) and 28,000 dying of AIDS-related illnesses annually [6]. Pharmacogenomic research involves studying how an individual's genes precisely influence their response to a given medication including drug efficacy, adverse events, and dosing requirements [7]. Pharmacogenomic research can present a vast amount of information [8, 9], some of which may be pleiotropic, where a single gene can contain information about several phenotypes, resulting into potential incidental findings [10, 11], in addition to primary results. Primary results are defined as findings responding to a well-defined research question, while incidental findings are any results discovered unintentionally and are not related to the primary research question, but may have significant implications for the health or well-being of the participant and family members [12].

Researchers and clinicians may use results from pharmacogenomic research analyses to determine the most appropriate drug and dosing requirements for an individual. Similarly, these results are equally important to research participants and communities involved in pharmacogenomic and other genomic research studies in determining their best treatment options. Studies have reported a high demand from research participants for their individual results from pharmacogenomic and genomic research [13, 14]. A recent study that explored the factors influencing the preferences and reasons for the desire to receive individual results from pharmacogenomic research among people living with HIV showed that 98% wanted to receive all their primary results [15]. However, several studies have reported participants' inadequate understanding of genomic and pharmacogenomic research information including results across the globe [16–20]. This has been attributed to the complexity of genomic terms and the absence of direct translations of genomic terms in many African local languages where genomic studies are being conducted [21–23]. A study conducted among PLHIV reported that only 23% of the participants enrolled in clinical trials that included a pharmacogenomic component had adequate understanding of the information disclosed to them during the consenting process [24].

Inadequate understanding of research information is a barrier to participants' informed decision-making for participation, and determining the kind of results they would like to receive. To promote adequate understanding of genomic and pharmacogenomic research information and findings, CE has been proposed by several researchers [25–28]. They suggested that communities should be actively engaged in research activities right from proposal development to result dissemination. CE is considered one of the platforms where research participants can learn more about how individuals' genes interact with drugs, an opportunity to clarify the research goal and objectives, and understand the appropriate CE approaches and strategies specific to a given research project [25, 29]. CE also facilitates co-learning between the researchers and research communities promoting transparency and free sharing, hence building long-term relationships based on mutual respect and trust [28, 30, 31]. The Uganda

National Council for Science and Technology (UNCST) guidelines for Community Engagement in Research recommend using one or more approaches, for example, formative consultations, existing structures and groups, community leaders, community events, mass media, and community advisory boards (CABs) when engaging research communities [29]. In addition, the H3Africa guidelines of community engagement for genomic research and biobanking in Africa recommend adopting responsive and flexible CE strategies that are shaped by the participants and their communities' experiences [25]. These guidelines emphasize that communities are not homogeneous, so researchers ought to employ creative approaches that minimize the potential risks to participants and research communities when communicating pharmacogenomic and genomic information. It is important to note that individual pharmacogenomic and genomic research results may present findings with ethical, legal and social implications that may not only affect research participants but also extend to families and research communities [32]. Therefore, inappropriate approaches of communicating pharmacogenomic research results at community and individual levels might raise concerns such as loss of privacy and breach of confidentiality, which may stigmatize research participants and their family members. Yet, there is limited literature on the suitable CE approaches for communicating simple and understandable pharmacogenomic information including results at community level. Therefore, this study explored stakeholders' perspectives on the role of CE in promoting understanding of pharmacogenomic research results among PLHIV. We present findings on CE approaches that can enhance participants' understanding of pharmacogenomic information and findings. We hope that findings from this study might inform institutional and national guidelines for returning genomic and genetic research results to people living with HIV, their families and communities.

## Materials and methods

### Study design and setting

This cross-sectional study used a qualitative exploratory approach [33–35]. The study was conducted at Makerere University College of Health Sciences (MakCHS) and five affiliated research institutions located on Mulago Hill. College of Health Sciences is one of the nine constituent colleges of Makerere University, with vast experience in HIV/AIDS research, including pharmacogenomics. We also considered three of the five accredited research ethics committees (RECs), housed at MakCHS, that had experience reviewing pharmacogenomic research specifically, for HIV treatment. In addition, we included three Ugandan RECs with prior experience in reviewing pharmacogenomic research in HIV.

### Research team

The research team comprised of a social scientist, bioethicists, a medical anthropologist, and medical scientists with experience in conducting and analyzing qualitative data.

### Study participants

In this study, we selected three categories of stakeholders in pharmacogenomic research for HIV treatment who were researchers, members of research ethics committees and community representatives. We purposively selected 15 researchers involved in pharmacogenomics research for HIV treatment for the period 2018–2021 and based at MakCHS or affiliate research institutes as eligible for enrolment in this study. Of these, 12 consented to participate, while three declined, citing inadequate time to participate in our study. We selected 12 REC members with experience in reviewing proposals for pharmacogenomic research in HIV

treatment. Six were REC chairpersons and the rest were members of the RECs representing the research communities. Three of the REC chairpersons preferred to nominate a member of their REC who was more knowledgeable in the field of study. We also conducted five deliberative focus group discussions (dFGDs), each comprising of six community representatives from the five HIV research institutions affiliated with MakCHS. This study was conducted between September 2021 and February 2022. All participants were purposively selected and were above 18 years of age. Before conducting the study interviews, the research team was trained on the protocol to ensure that they understood the study well.

## Study procedure

Considering the restrictions that prevailed during the COVID-19 pandemic, some data collection activities were conducted virtually. Researchers and REC members who were involved in or had reviewed HIV pharmacogenomic research respectively, were contacted by email, which contained a brief description of the study and a request to schedule an appointment for the interview. Expression of interest to participate was recorded by a positive response to the email, followed by sharing the consent form. The consent forms were either signed electronically or by a written email notification of acceptance to participate in this study. Appointments were scheduled and conducted virtually via Zoom at the participants' convenience. Interviews were audio-recorded and lasted between 30 and 40 minutes.

Five dFGDs were conducted with community representatives. The community representatives were contacted through their respective leaders. The dFGDs were conducted in two languages, English and Luganda, the commonly spoken language in Central Uganda. The dFGDs were conducted through face-to-face interactions after the national lockdown was lifted during the COVID-19 pandemic, with appropriate mitigation measures in place. Prior to the dFGDs, each community representative received a consent form document in the language of their choice. One of the research team members obtained written consent from each community representative after discussions on the various components of the study. At the beginning of each dFGD, participants were provided with an overview of how antiretroviral drugs interact with human genes. This was followed by a vignette describing a hypothetical scenario of the possible results that could be generated from pharmacogenomic research, and these included primary results and incidental findings. Prior information helped participants to gain an understanding of pharmacogenomic research and the different kinds of results that could emerge from it. Clarifications were offered prior to and during the discussions. A team of four researchers (SN, AT, CW and ESM) conducted all interviews to ensure consistency. A note-taker was present throughout the discussions to back up the electronic recording. The dFGDs were audio-recorded and lasted about 60–90 minutes. Both dFGD and interview guides were developed from the literature [36–43] and subsequently revised to capture new emerging topics. The topics of discussion included questions related to the potential benefits of community engagement, how to effectively communicate pharmacogenomic research information including primary results and incidental findings at community level, how to achieve community consensus when determining the kind of results to be returned, perceived challenges of engaging communities in the feedback process of results, and the perceived roles of each stakeholder in promoting research participants' understanding of pharmacogenomic research results. The guides were first piloted on three genomic researchers, two REC members and three HIV peer support members who were excluded from the study. However, their feedback was used to improve the interview guides. The research team held debriefing meetings at the end of each interview to identify new perspectives that were not initially captured by the tool. Data were collected until no new information or insights were being revealed.

## Data analysis

Data were analysed continuously throughout the study using a thematic approach [44, 45]. All audio recordings were transcribed verbatim. Transcripts of the community representatives were translated from Luganda to English. All transcripts were verified for accuracy by reading word by word while listening to the audio recordings for quality checks and spelling errors. This step helped the authors to familiarize, mark and memo the data. Transcripts were initially analyzed separately based on the three participant categories: researchers, REC members and community representatives, in order to gain a deeper understanding of each category's perceptions on the subject matter. Three authors (SN, AT, and ESM) selected three transcripts from each participant category for open coding. These scripts were read line by line to generate the first set of codes. Synthesis of codes from the independent reading were iteratively discussed among the three authors and codes of similar ideas were merged. Differences in coding among the independent coders were resolved by consensus. A hierarchal codebook and coding framework was developed for each participant category by the three authors to guide the analysis of the data. The hierarchy of codes was then sorted into categories based on how themes were related and linked. We then deductively generated themes using our pre-existing analytic framework, which we developed from the literature on the role of community engagement in genomic research, as represented in the interview guides. We also inductively considered new themes that merged from the transcripts. All the transcripts were then imported into Nvivo version 12 [46] and coded by three authors (SN, AT, and EMS). Three authors DK, IM and CW examined the themes for patterns consistency until consensus was achieved on the final themes. All the authors compared the emergent themes with the existing literature to confirm that the final themes accurately represented the stakeholders' perspectives on the role of community engagement in promoting participants' understanding of pharmacogenomic research results. We also returned some transripts to the stakeholders to verify whether the data collected was a true reflection of their statements on the subject matter. This ensured that the data can be transferable to similar settings and enhanced the credibility of the study findings [25, 47]. The key findings were summarized and the overlapping themes across the three categories of participants were merged as presented in Table 2. The final codebook included the merged themes and codes from all three categories of participants. It was continuously refined to establish the themes presented in the results section.

Regarding research reflexivity, the research team was aware that we needed to remain neutral throughout the interviews and focus group discussions. We acknowledge our potential biases based on the prior knowledge about the research institutions where we recruited the research stakeholders and the existing relationships between the interviewees and the research team through prioritizing listening from the interviewees' perspective.

## Ethical consideration

This study obtained ethics approval from the Makerere University School of Biomedical Sciences Higher Degrees and Research Ethics Committee (SBS- 855) and Uganda National Council for Science and Technology (SS 735ES). Written informed consent was obtained from all stakeholders, and they were assured of confidentiality.

## Results

The demographic characteristics of stakeholders are presented in Table 1. A total of 54 participants participated in this study. The majority had more than five years' experience in pharmacogenomic research and HIV care and treatment.

**Table 1. Stakeholders' demographic characteristics.**

| Participants | Researchers | REC members | Community Representatives |
|---|---|---|---|
| N = 54 | 12 | 12 | 30 |
| **Sex** | | | |
| Men | 07 | 05 | 16 |
| Women | 05 | 07 | 14 |
| **Highest level of Education** | | | |
| Primary | | | 02 |
| Secondary | | | 07 |
| Diploma | | | 13 |
| Bachelor | 02 | | 07 |
| Master | 08 | 07 | 01 |
| PhD | 02 | 05 | |
| *Research Experience in PGx research and HIV treatment and care | | | |
| < 5 years | 02 | 02 | 07 |
| 5–10 years | 06 | 04 | 12 |
| >10 years | 04 | 06 | 11 |

*PGx stands for Pharmacogenomic research

## Summary of the themes and key findings

A summary of the key themes that emerged from the data collected as described in Table 2.

  I. Benefits of engaging communities prior to returning individual pharmacogenomic research results

 II. Obtaining community consensus on the kinds of pharmacogenomic results to be returned.

III. Opinions on how pharmacogenomic research information and results should be communicated at community and individual levels

IV. Perceived roles of community stakeholders in promoting participants' understanding and utilization of pharmacogenomic research results

 V. Perceived challenges of engaging communities when returning individual results to research participants

**Benefits of engaging communities prior to returning individual pharmacogenomic research results to participants.** Majority of the stakeholders (47) lauded CE for promoting understanding of individual pharmacogenomic results, but emphasized the need to engage community members right from proposal development. Stakeholders considered CE an opportunity to create platforms where participants and community members can freely interact with researchers throughout the research period. In addition, engaging communities was recognized for enabling researchers to learn about communities' cultural values and local explanations of disease experiences. In return, participants and communities also learn about the individual variability of drug response, thus promoting adequate understanding of pharmacogenomic research and the implications of research results. The continuous interactions between researchers, participants and communities were also said to foster lasting relationships based on mutual respect and trust, and eliminate misconceptions about genomics research.

**Table 2. Emergent themes.**

| Theme | A summary of findings from various stakeholders |
|---|---|
| 1. Benefits of engaging communities prior to returning individual pharmacogenomic research results | • Facilitates cross-learning between researchers and community members.<br>• Fosters long-term relationships between researchers and community members.<br>• Promotes ownership of results and solidarity.<br>• Overcomes societal stigma and misconceptions. |
| 2. Obtaining community consensus on the kinds of pharmacogenomic results to be returned. | • First, define the community with a likelihood to benefit from these results.<br>• Provide generalized and understandable information about pharmacogenomic research, kinds of results that can be returned and their ethical, legal and social implications.<br>• The need to balance community and individual interests.<br>• Obtaining consent from family members before sharing information and results of pharmacogenomic research. |
| 3. Opinions on how pharmacogenomic information and research results should be communicated at community and individual levels. | • Using a language preferred by the community members<br>• Employing a genetic counselor to aid the research team in determining the appropriate communication strategies and tools.<br>• Training peer-clients with skills in explaining and communicating certain genomic terms to research communities.<br>• Using institutional music and drama clubs to communicate pharmacogenomic research information and results<br>• Group education to promote participants' learning from each other |

(*Continued*)

**Table 2.** (Continued)

| Theme | A summary of findings from various stakeholders |
|---|---|
| 4. Perceived roles of community stakeholders in promoting participants' understanding and utilization of pharmacogenomic research results. | • *Researchers:*<br>  • Provide basic, adequate, and understandable information,<br>  • Ought to be knowledgeable about the implications of the results,<br>  • Encourage public sensitization and community education about the role of genes in the human body.<br>  • Provide a well-explained action plan after returning results, and<br>  • Report study outcomes to RECs & regulators.<br>• *Community members:*<br>  • Identifies potential social harms from the results,<br>  • Use familiar/ relatable life stories to facilitate understanding of genomic terms,<br>  • Provide accurate information to overcome societal misconceptions.<br>• *REC members:*<br>  • Identify potential ethical issues that may arise from returning results,<br>  • Ensure that information provided to participants and communities is sufficient and understandable.<br>• *Research participants:*<br>  • Raise questions and concerns about unclear concepts for clarification,<br>  • Express their interest either to receive or not to receive results.<br>• *Research institutions:*<br>  • Put measures in place to protect participants and research communities from possible social harm after learning about these results.<br>  • Facilitate a smooth referral to health facilities offering various genetic services.<br>• *National research regulators:*<br>  • Develop contextualized guidelines for returning individual and aggregate genomic research results.<br>  • Develop a framework to address the need to balance community interests and individual interests when selecting the kinds of results to be returned to research communities. |
| 5. Perceived challenges of engaging communities when returning individual results to research participants. | • Difficult and time consuming to determine the most suitable CE approach and language to use when interacting with multilingual communities.<br>• Fear of losing the scientific meaning when translating genomic information into drama and songs.<br>• Absence of genetic counselors in Uganda. |

*[. . .. . .]As long as you keep giving information to people about unclear things. . .they can easily trust you. This will help reduce on many community misunderstandings and misconceptions about genomic research.*

(KII_Male_Researcher_12)

*You see, you need to engage the communities early enough. When you do so, many will be given an opportunity to understand the goal of pharmacogenomic research and its implications even if they may not necessarily participate, but they will have learnt about these genetics hard concepts. So here, you see, these results do not necessarily help the participants alone but the whole community who have very many questions about this topic. . .*

(FGD 2_Male_Community representative 3)

One community representative mentioned that community engagement can promote ownership of findings of pharmacogenomic research and foster community solidarity, where community members may contribute to the financial and social support for the less privileged individuals.

*When they receive the research information and results as a family or as community, they will be more willing to learn more about genes and drug interactions. . .and then take necessary action. I have even seen sometimes, people in communities come together to collect finances for their friends to go abroad for expensive medications which they [patients] cannot afford on their own.*

(FGD 1_Female_Community representative 4)

Majority of the stakeholders across the different participant categories (38) acknowledged community engagement as an important strategy to overcome societal stigmatization of individuals or families who may not respond well to the available HIV treatment regimen provided by the Government of Uganda.

*Most of the time people in communities believe that individuals or families that are different from the majority of the population are cursed or are being punished by God. But when you discuss these [genetics] issues together with them [community members] as a group, many will appreciate the importance of having a genetically diverse population*

(KII_Male_Researcher_6).

**Obtaining community consensus on the kinds of pharmacogenomic results to be returned.**   We asked stakeholders about the various ways of obtaining community consensus on the kinds of results they would wish to learn from pharmacogenomic research. All stakeholders across the different participant categories (54) agreed that a community is a complex group of people with varying values, beliefs and interests. Therefore, stakeholders emphasized the need to first define the specific groups of people within the community that have potential to benefit from the genomic research study, and later, identify the key gatekeepers prior to accessing the potential beneficiaries.

*I think in this case, it would be important to first define a community based on the disease and subject at hand, if we want them [communities] to appreciate the benefits of research. Research targets a particular population and not the whole community, so we should be able to engage with people who have been infected and/ or affected by HIV/AIDS and see how these results can be returned them.*

(KII_Female_Researcher_5).

Almost all of the stakeholders (50) agreed that before selecting the kinds of results from pharmacogenomic research, community members should be provided with adequate information about the nature of the results and their implications. One REC member emphasized that researchers and community representatives should provide generalized but appropriate and understandable information about the role of genes in the treatment of HIV/AIDS to minimize raising ethical issues in communities, such as stigmatization.

*We need to strike a balance regarding the information researchers communicate at community level and individual level. People living with HIV still suffer stigmatization, even among themselves. Just telling someone that your body cannot respond to first-line ARVs [Antiretroviral drugs], you need to move to second line may make one feel stigmatized. So, I feel we should provide general information about pharmacogenomic research at community level and then get into the specifics at individual level.*

(IDI_Male_REC member_5).

Another REC member cited concerns about balancing community and individual interests when selecting the kinds of results to be returned from pharmacogenomic analyses. Since community choices are made based on a majority decision, sometimes the voices of the minority may be ignored, yet individuals' rights should be respected. He suggested that research institutions and regulators should develop a framework for building community consensus on the kind of results to be discussed at community level while considering the interests of the individuals participating in pharmacogenomic research.

*. . ... As communities are pushing to be involved in the process of returning these results, we need a framework and guidance from regulators and RECs on how to balance participants' interests at individual level and communities' interests at community level.*

(KII_Male REC member #12)

Several REC members (06) emphasized the need to seek consent from the family head and individual consent from each family member who wishes to learn about their pharmacogenomic research results. They emphasized the need to provide adequate information and time to the family members to understand the results' implications fully, before soliciting their choices of the kinds of results they wish to receive.

*These family members should be first prepared by providing them with detailed information about genes and their role in their bodies just like how you have prepared the participant throughout the study. In fact, these people [family members] should first consent to either accept to know this information or not. Some family members may not want to know about these results. No one should decide for them, not even the participant's decision to share his/ her results with family members. They [family members] should also be autonomous because the results will be affecting their lives as individuals. . ...*

(KII_Male_REC member # 6)

**Opinions on how pharmacogenomic research information and results should be communicated at community and individual level.**   In order to build community trust and promote ownership of these results, stakeholders suggested that a healthcare provider knowledgeable and experienced in genetics should provide the pharmacogenomic information

and results to communities. Eight community representatives emphasized the need to provide genomic information in local languages that local communities can understand.

*The language researchers use to communicate this information matters. Researchers should consider translating this information into local languages and consult community representatives on the appropriate local words to be used when explaining genomic terms.*

(FGD 5_Male_Community representative 5).

Majority of the researchers (10) emphasized the role of genetic counselors in the feedback process of pharmacogenomic research results. One researcher mentioned that genetic counselors should work with the research team to devise appropriate tools for communicating pharmacogenomic information and results, and guide on the strategies for engaging communities throughout the research project.

*Sharing this genetics information with community members can sometimes be tricky. Many of them [community members] are illiterate and do not know much about genes. Therefore, the research team needs to be very creative and probably work with genetic counselors who understand both the scientific information and societal norms.*

(KII_Female_Researcher_2)

Another researcher mentioned that the research team usually trains peer clients to communicate pharmacogenomic research information to fellow participants. She hoped this strategy would enhance participants' understanding of individual pharmacogenomic research results, since some participants may be more willing to open up to their peers than healthcare workers.

*I think we can also utilize our peer-clients to explain these results to their fellow participants. From the time we started these PG [pharmacogenomic research] studies, we have trained our expert patients how to explain and break down these terms. They even share with us some of the concerns that some participants tell them. . ..*

(KII_Female_Researcher_8)

Some researchers (04) suggested using music, dance and drama to convey information about pharmacogenomic research and the implications of its results to participants.

*Another way we can communicate these results to participants and community members is utilizing the creativity of our drama club here at the Institute. They usually come up with skits and songs to communicate HIV-related information at least once in two weeks. So we can give them scripts to act out something about the role of genes in breaking down the ARVs [Antiretroviral drugs].*

(KII_Male_Researcher_11)

One researcher suggested using group education as another way of promoting participants' understanding of pharmacogenomic research information including results.

*I would think about holding group discussions among the potential participants who may be attending the ART [Antiretroviral treatment] clinic on a given day. For example, I could plan*

*a session on one Tuesday, the day when many patients come in, then I give them some information about the study and encourage them to ask questions. This way, some other potential participants can learn from others' questions but can ask more questions when they choose to join the study*

(KII_Female_Researcher _2)

One community representative emphasized the need to identify a family elder or an influential member of a particular family to communicate these results at the family level.

*If you want participants and families to own and utilize these results, you may need to identify a key person in that family, someone that is respected and can be listened to by most family members. . .. If one of their own can describe a given condition, while citing familiar stories, it will be easy for other family members to appreciate the results and own them. . .*

(FGD 1_Male_Community representative 5).

**Perceived roles of community stakeholders in promoting participants' understanding and utilization of pharmacogenomic research results.** Stakeholders made several suggestions on what they perceived as the roles of the different stakeholders in promoting participants' understanding and utilization of pharmacogenomic research results. They highlighted the contribution of participants, community representatives, researchers, research institutions and research regulators to enhancing community understanding of pharmacogenomic research results.

*Researchers* asserted that adequate and relevant information about pharmacogenomic research should be clearly provided to research participants and communities throughout the study. However, one researcher emphasized the need to avoid redundant information that may not have meaning to research participants and the community members. Furthermore, they indicated that researchers ought to be knowledgeable and competent to interpret pharmacogenomic results, fully understand their implications and provide a well-explained action plan following the return of results to participants and their communities. Two researchers mentioned that it is their responsibility to share study findings with the national regulatory agencies and RECs to maximize utilization of research findings. They emphasized working together with the Ministry of Health to educate the public about the different roles of genes in the human body.

*"What we [researchers] should do is to understand the implications of these results first, before presenting them to the participants, or to their family members. This way, we shall be able to provide answers and appropriate solutions to participants and their family members.*

(KII_Male_Researcher_6)

*"We have the moral obligation to share all these findings with the Regulatory Authorities and ethics committees, whether primary results information or incidental findings. This information can help guide policy makers on what they may think is crucial to consider about pharmacogenomics in HIV treatment or other genomic studies.*

(KII_Male_Researcher_12)

*Community representatives* indicated that its is their responsibility to guide the research team when assessing the potential social harm and other risks of returning individual

pharmacogenomic research results to participants, their families, and community members. They mentioned that it was their responsibility to enhance community members' understanding of research findings by explaining complex genomic terms using relatable and relevant life stories. They also felt that it was their responsibility to create awareness about pharmacogenomic research and the implications of research findings through sensitization and health campaigns. They argued that health campaigns would help to provide accurate information about pharmacogenomic research and minimize misinformation and misperceptions.

*. . .. You know genetics is a very sensitive topic and many communities have not appreciated these studies right now. There are many incorrect stories and myths about HIV drugs that come up because people respond differently to the ARVs [Antiretroviral drugs]. . . And some people take their ARVs [Antiretroviral drugs] along with some traditional herbs, which may also affect how their bodies react to the ARVs [Antiretroviral drugs]. . . so the community representatives should come out and provide accurate information to nullify the myths about how different human bodies respond to ARVs [Antiretroviral drugs]. . ..*

(FGD 5_Female_Community representative 1)

REC members highlighted some of their responsibilities in promoting participants' understanding of pharmacogenomic research results. They stated that it is their role to review research protocols and identify the possible ethical, social, and legal implications of sharing individual pharmacogenomic research results with research participants and their communities. They also pointed out the importance of ensuring that the informed consent documents have accurate information, are simple and written in a language that is easily understandable. They further suggested that researchers should put in place measures to protect participants and families from social harms that may be associated with the results feedback process. REC members also emphasized the need for researchers to submit study-specific dissemination plans for review and approval prior to the commencement of study activities.

*".. The REC's primary role is to protect participants and their communities from research related harm. . ... for example we need to ensure that the information given to them is simple, clear and easy to understand because the results can be misinterpreted and cause psychological harm to participants. . .*

(KII_Male_REC Participant #10)

In addition, RECs should provide standardized protocol templates for genetic and genomic studies to guide researchers on the critical ethical aspects of genomic and genetics studies when developing research protocols.

*RECs have a role of reviewing and advising investigators how to design informed consent processes and how these results can be safely returned. We can achieve this by standardizing genetics protocol templates for genetics and genomic studies..*

(KII_Male_REC Participant #12)

Regarding the role of *research participants and community members*, stakeholders said that research participants and members of research communities should raise questions and concerns about unclear concepts for clarification during their interactions with research teams. They said that research participants have a role of selecting the kind of results they would wish to receive after full comprehension of the implications of their choices.

*We try our level best to build a good relationship with our participants so that they can freely and openly tell us their concerns and aspects they might not have understood. So we expect them to ask questions and also tell us truthful information during our discussions with them*

(KII_Female_Researcher_3)

*I think research participants have the responsibility to clearly inform the researchers whether they want to receive their results or not. They should also specify the kind of results they would like to receive. They should also let us know whether it is okay to share their results with family members or not*

(KII_Male_Researcher_10)

Stakeholders asserted that *research institutions* should put in place measures to protect research participants and communities from possible social harm that may arise from the result feedback process.

*Research institutions should come up with structures and systems that protect participants and communities from possible harm. These institutions should employ full-time genetic counselors to address genetics related questions and concerns to both individual participants and communities, even when the research project has ended.*

(IDI_Female_REC member_6)

Stakeholders also felt that research institutions should encourage continuous feedback from research participants and communities even after the study closure, for example using suggestion boxes at the facilities. Further, stakeholders indicated that institutions should engage and /or collaborate with health facilities offering genetic services for a smooth referral of participants and building new relationships at these facilities.

*One thing I would request research institutions to do, is to establish collaborations with other health facilities or NGOs that can provide extra support to our participants and their family members after we share these results. You may find that some participants need extra psychological support than just the counseling services we offer here [HIV/AIDS research clinics]*

(KII_Female_Researcher_5)

Regarding the role of *national research regulators* for example the Ministry of Health, UNCST and NDA, stakeholders asserted that the regulators should amplify the need to provide individual results of genomic analyses particularly those that are of clinical significance. Stakeholders also suggested that the national research regulators should develop contextualized guidelines to facilitate a safe return of individual genomic and genetic research results to participants, family members, and research communities.

*We currently do not have national guidelines on returning genomic and genetics research results to participants and community members. . .. Therefore, it is sometimes hard for us REC members to advise researchers on some ethical aspects that may affect participants in any way. So our regulators need to develop these guidelines as soon as possible*

(IDI_Male_REC member_12)

**Perceived challenges of engaging communities when returning individual results to research participants.** Two researchers observed that it might be difficult to determine the most suitable CE approach when engaging community members. They advised that sometimes, the research team needs to be flexible to accommodate more than one approach. This may require a lot of time and financial resources to creatively adapt more than one approach when engaging and communicating pharmacogenomic related -information appropriately without losing scientific meaning.

*[. . ..] pharmacogenomic research is already difficult to understand. So the team [research team members] need to be careful when translating this information in plays and songs not to lose the scientific meaning of this research [pharmacogenomic research]. . ..*

(KII_Female_Researcher_8)

*[. . ..] Sometimes deciding on what language to use when speaking to a group of people is very hard. In Kampala, people speak more than three common languages so the researchers may even get confused on the most suitable language to choose over others.*

(FGD 1_Female_Community representative 4)

One third of the community representatives raised concerns about diagnostic misconceptions of genetics and genomics studies.

*Right now, there is a lot of 'noise' about paternity testing in our communities. Even many labs (laboratories) are making adverts for people to go for paternity testing. Now, when you talk about anything concerning genes, many people's minds run to testing their children's paternity. . ..*

(FGD 5_Female_Community representative 1)

One REC member said that prioritizing certain kinds of results over others could be challenging to communicate at community level compared to individual level. He felt that it is sometimes difficult to give an answer regarding why emphasis is placed on certain kinds of results while leaving out others.

*[.. . ..] I can see it being difficult to explain why researchers are concentrating on certain kinds of results and missing others [results]. They [participants] might think that other results are not important. And, you see, our people are shy. They will fear to ask questions in a group of people.*

(KII_Male REC member #12)

Many researchers (08) were concerned about the absence of genetic counselors at their institutions who are skilled in communicating genomic/ genetic information while respecting the community values and beliefs.

*I worry about if we [researchers] can communicate these results to our participants effectively. We don't have genetics counselors in my institution. . ...And I think we may ignore some ethical and social challenges that may affect our people.*

(KII_Male_Researcher_6)

## Discussion

This study explored the role of community engagement in promoting understanding of individual pharmacogenomic research results among PLHIV. Pharmacogenomic research is paving way for the future of personalized medicine [48, 49], where tailor-made treatment strategies are defined for groups of individuals based on their genetic make-up. Results from pharmacogenomic research may be used to guide clinicians and researchers' decisions on the appropriate drug and drug dosage required to produce a desirable effect for an individual or groups of people based on their genetic makeup [4]. Similarly, returning these results to PLHIV could help them understand why some people respond to the same drugs differently from others. Participants' understanding of their genetic make-up encourages adherence to the prescribed treatment, especially among PLHIV with long treatment periods [50, 51]. However, it is imperative that PLHIV attain full understanding of the ethical, legal and social implications before receiving individual results of pharmacogenomic research. This is because these results can lead to social and psychological harm to participants and their family members [32]. Studies have reported the complex nature of genomic terms that are difficult to understand even among the literate populations, low literacy levels in many resource limited settings, and community misconception of linking participation and receiving results from genomic and genetic results to establishing the paternity of children and other family members as contributing factors to participants' inadequate understanding of pharmacogenomic and genomic information [32, 52–54]. To overcome these challenges, it is important that communities likely to benefit from a genetic or genomic study are actively engaged throughout the study. Respondents agreed that determining a culturally acceptable approach or approaches is an essential step to achieve effective CE. Research teams should select an approach that is creative, flexible and sensitive to participants' values, beliefs and education levels. Their views are consistent with a study that explored perspectives on returning individual and aggregate genomic research results to study participants and communities in Kenya [55]. Researchers suggested using music, dance and drama as an approach that could enhance communication of understandable information about pharmacogenomic research and results at community level. A study conducted in South Africa used the drama of DNA approach to engage communities reported that drama can be a relatively effective approach in engaging community members when conveying information about ethical and social challenges related to the return of individual genetic research results [56]. Music and drama has been used to convey information related to HIV prevention, and adherence to ARV treatment in Uganda and many parts of sub-Saharan Africa [57–60]. Therefore, research teams might adapt the music and drama approach to communicate understandable information about the implications of findings from pharmacogenomic research since communities of PLHIV are familiar with. However, caution is necessary during the development of music and drama scripts and translation of research information from English to local languages. This is to avoid losing the scientific meaning of how genes interact with medicines. Therefore, the research teams should work together with drama teams in the development of drama scripts to ensure communication of accurate information in a simple manner that is understandable by community members.

Respect for individual's privacy and confidentiality should be paramount when communicating information about pharmacogenomic research results through music and drama at community level. Upholding participants' privacy and confidentiality when communicating results from genomic and pharmacogenomic research prevents risks of stigmatizing participants and their families, discrimination, and lack of interest in participating in future genomic and genetic research [61–64]. In order to maintain participants' privacy and confidentitality, stakeholders mentioned that relatively general but understandable information about pharmacogenomic

research results should be provided at the community level, while providing individualized information about the results to participants during one-on-one discussions with the participants.

Researchers also suggested training peer clients to provide additional explanation to participants on how an individual's genes interact with drugs in a layman's language. Empowering community members as vessels to explain genomic and genetics to their fellow peers might encourage free and open sharing of feelings about these results. Trained peer-clients also provide social support to fellow peers to overcome fears and misconceptions about findings from genomic and pharmacogenomic research, hence promoting participants' understanding and ownership of the results. Group education is also another approach that was suggested by researchers to promote participants' understanding. This approach might provide an opportunity for potential participants to ask questions based on the background information given during the group discussions thus enabling their fellow peers to learn from the explanations provided by the researchers. It is important to note that participants who might not be comfortable raising certain questions during the group discussion are still able to ask their questions during their individual meetings with the research teams.

Respondents also suggested the need to engage genetic counselors when determining an appropriate communication method of returning these results to communities. Currently, there are few genetic counselors in some sub-Saharan African countries while others do not have any genetic couseller, yet genomic research in rapidly increasing in Sub-Saharan Africa [65, 66]. Genetic counselors are skilled professionals in providing scientific information about genomic and genetics and social support to participants involved in genomic research. Therefore, research institutions should develop capacity of genetic counselors to support the process of returning individual pharmacogenomic research results.

Respondents also suggested that research insitutions should hold regular and ongoing discussions about genomic and genetic research. The institutions may develop a consistent schedule for discussions about genomic research as an opportunity for participants to appreciate the relevance of genes in an individual's body and implications of receiving individual pharmacogenomic research results. However, some research institutions in developing countries might face challenges with limited funding to achieve effective community engagement. Therefore, research institutions may solicit financial support from the government and non-government agencies to achieve effective community engagement.

Lastly, key stakeholders for example researchers, REC members, and community representatives involved in pharmacogenomic research have a role in ensuring that participants adequately understand the implication of genomic and pharmacogenomic research results before they are returned to them. Researchers should report study findings to the national research regulators and policy-makers to jointly develop appropriate strategies of sensitizing communities about the various roles of genes in the human body, thus promoting participants' understanding of geneomic and pharmacogenomic research. Researchers, together with other stakeholders should protect participants and research communities from possible harm that might arise from returning individual pharmacogenomic research results to participants. In addition, researchers should sensitize communities about the various functions of genes in an individual's body. This may help community members to overcome misconceptions about of receiving results from genomic and pharmacogenomic research studies. Similarly, community representatives should also encourage sharing of correct information about genomic research to demistify the existing diagnostic misconceptions in communities. Community representatives and research participants should raise concerns or questions on unclear information about genomic and pharmacogenomic research and the implications of the results on their lives. Research regulators should develop guidelines and frameworks that facilitate adequate understanding of genomic and pharmacogenomic results at the individual and community levels.

Our study had some limititations. Some interviews were conducted virtually via Zoom due to the COVID-19 pandemic, whose mitigation measures restricted face-to-face interactions. Thus, the authors were not able to capture non-verbal communication for some respondents. Sometimes, the research team experienced challenges with network connectivity. However, zoom interviews were substituted with telephone calls and respondents were encouraged to share additional information via email after the interviews.

## Conclusion

Our findings show that there is a consensus among the different stakeholders that CE can play a vital role in promoting research participants' understanding of individual pharmacogenomic research results. Respondents mentioned several CE approaches including adapting existing music, dance and drama clubs, group education and training peer clients to communicate understandable information about pharmacogenomic research and the implications of its results to research communities. However, these approaches should comply to the ethical standards of conducting research such as respect of participants' privacy and confidentiality. We recommend further research to explore the feasibility of using the existing CE approaches to communicate simple and understandable information and the implications of the results to research participants at community level. Of concern, respondents emphasized the need to engage genetic counselors when determining the suitable approach or approaches to achieve meaningful community engagement, yet many research instititions conducting genomic research do not have genetic counselors. We recommend building capacity for genetic counselors in Uganda and other sub-Saharan countries where genomic and genetics research is conducted. We also recommend developing a framework that respects individuals and community interests, values and literacy levels when communicating the pharmacogenomic research information and results at individual and community levels.

## Supporting information

**S1 Dataset.**
(DOCX)

**S2 Dataset.**
(DOCX)

**S3 Dataset.**
(DOCX)

## Acknowledgments

The authors would like to acknowledge the assistance offered by the Faculty at Berman Institute of Bioethics, Johns Hopkins University and the Faculty of Makerere University International Bioethics who contributed to the study design and interpretation of the results.

## Author Contributions

**Conceptualization:** Sylvia Nabukenya, David Kyaddondo, Ian Guyton Munabi, Catriona Waitt, Erisa S. Mwaka.

**Data curation:** Sylvia Nabukenya, David Kyaddondo, Ian Guyton Munabi, Adelline Twimukye, Erisa S. Mwaka.

**Formal analysis:** Sylvia Nabukenya, David Kyaddondo, Ian Guyton Munabi, Catriona Waitt, Adelline Twimukye, Erisa S. Mwaka.

**Methodology:** Sylvia Nabukenya, Ian Guyton Munabi, Adelline Twimukye, Erisa S. Mwaka.

**Supervision:** David Kyaddondo, Erisa S. Mwaka.

**Validation:** Catriona Waitt, Erisa S. Mwaka.

**Writing – original draft:** Sylvia Nabukenya.

**Writing – review & editing:** Sylvia Nabukenya, David Kyaddondo, Ian Guyton Munabi, Catriona Waitt, Adelline Twimukye, Erisa S. Mwaka.

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
