## [Decision Letter · Decision Letter 0]

11 Oct 2023

PONE-D-23-22714The role of community engagement in promoting research participants’ understanding of pharmacogenomic research results: Perspectives of stakeholders involved in HIV/AIDS research and treatmentPLOS ONE

Dear Dr. Nabukenya,

Thank you for submitting your manuscript to PLOS ONE. After careful consideration, we feel that it has merit but does not fully meet PLOS ONE’s publication criteria as it currently stands. Therefore, we invite you to submit a revised version of the manuscript that addresses the points raised during the review process.

hank you for the opportunity to review your study.  We see this as an important area of investigation for dealing with a major pandemic threat. in this regard the manuscript was reviewed by an expert in the area who has provided comprehensive feedback, to which l concur, on a number of issues that will need to be addressed if the manuscript will be accepted for publishing at PLOS ONE. If you choose to resubmit we request that you carefully note the comments by the reviewer below, and address each issue in your resubmission. Alternatively, you may consider submitting to another journal. If you choose to resubmit, please submit your revised manuscript by Nov 25 2023 11:59PM. If you will need more time than this to complete your revisions, please reply to this message or contact the journal office at plosone@plos.org. Please include the following items when submitting your revised manuscript:A rebuttal letter that responds to each point raised by the academic editor and reviewer(s). You should upload this letter as a separate file labeled 'Response to Reviewers'.A marked-up copy of your manuscript that highlights changes made to the original version. You should upload this as a separate file labeled 'Revised Manuscript with Track Changes'.An unmarked version of your revised paper without tracked changes. You should upload this as a separate file labeled 'Manuscript'.

We look forward to receiving your revised manuscript.

Kind regards,

Tahir Turk, PhD

Academic Editor

PLOS ONE

Journal Requirements:

"This study was funded by the Fogarty International Center of the National Institute of Health through the Makerere University International Bioethics Research Training Program Grant Number D43TW010892. The authors would like to acknowledge the assistance offered by the Faculty at Berman Institute of Bioethics, Johns Hopkins University and the Faculty of Makerere University International Bioethics who contributed to the study design and interpretation of the results. Waitt C is funded by Wellcome Trust Clinical Research Career Development Fellowship. The content is solely the responsibility of the authors and does not necessarily represent the official views of the National Institutes of Health."

"This study was funded by the Fogarty International Center of the National Institute of Health through the Makerere University International Bioethics Research Training Program Grant Number D43TW010892. The funders had no role in study design, data collection and analysis, decision to publish, or preparation of the manuscript."

Reviewers' comments:

Reviewer's Responses to Questions

**Comments to the Author**

1. Is the manuscript technically sound, and do the data support the conclusions?

Reviewer #1: Yes

2. Has the statistical analysis been performed appropriately and rigorously? 

Reviewer #1: N/A

3. Have the authors made all data underlying the findings in their manuscript fully available?

Reviewer #1: No

4. Is the manuscript presented in an intelligible fashion and written in standard English?

Reviewer #1: Yes

5. Review Comments to the Author

Reviewer #1: Thank you for the opportunity to review your study. I welcome more literature to bolster Community Engagement practices in different parts of public health and the HIV response.

The study design is adequate, as are the data collection and analysis sections. I do have some recommendations that might increase the study's impact.

#1) Differentiate with responses. When I conduct focus groups, I usually have a design that differentiates people's roles to different focus groups. This way I can look at comparisons across groups with norms and concerns. You've presented the data as if all respondents - researchers, ethics committee members, and community members - all agree across every issue.

#2) Identify what is important in the study and lean into it. You're providing stories on how various actors feel about concepts of community engagement. The report back on findings seems dry, and basically is reduced to "Community Engagement is important." Why and how are key questions that you do not approach directly.

#3) Avoid jargon. While you discuss pharmacogenomic research, it is often jargon. What makes this kind of research important for people living with HIV and who decides it's important? What are the ramifications of the research? There are times when you seem to conflate research with clinical practice (e.g. the individual wants their data from research). How is this possible? You make an assumption that all readers are up to date in this part of the field.

#4) Barriers to Community Engagement are not identified. This is odd, as most researchers will discuss their frustrations openly (e.g. funding requirements, time, lack of expertise) yet there was little of that here (outside of language). Other barriers could be highlighted more.

Thank you for the opportunity to review your study.

6. PLOS authors have the option to publish the peer review history of their article (what does this mean?). If published, this will include your full peer review and any attached files.

Reviewer #1: No

---

## [Author Response · Author response to Decision Letter 0]

3 Dec 2023

22 November 2023

Academic Editor,

PLOS ONE Journal

Dear Dr Tahir Turk,

RE: Response to reviewers’ comments.

On behalf of the authors, I take this opportunity to appreciate you and the reviewers for sparing time to review and provide insightful feedback to improve the quality of our manuscript entitled “The role of community engagement in promoting research participants’ understanding of pharmacogenomic research results: Perspectives of stakeholders involved in HIV/AIDS research and treatment” PONE-D-23-22714 for consideration by the PLOS ONE journal.

I hereby submit a point-by-point response to the peer review comments of the above named manuscript (attached).

We have positively responded to all the concerns raised by the reviewers and we believe that their feedback has greatly improved the quality of our work and readability of our manuscript.

Yours sincerely,

Sylvia Nabukenya

Email: nabukenyas89@gmail.com

RESPONSE TO ADDITIONAL JOURNAL REQUIREMENTS

ID COMMENTS RESPONSES

01 Please ensure that your manuscript meets PLOS ONE's style requirements, including those for file naming. The PLOS ONE style templates can be found at 

Thank you very much for raising this comment and providing the links to the style templates. We have revised the format of the manuscript as guided.

02 We suggest you thoroughly copyedit your manuscript for language usage, spelling, and grammar. If you do not know anyone who can help you do this, you may wish to consider employing a professional scientific editing service.

 The name of the colleague or the details of the professional service that edited your manuscript Thank you very much for raising this comment. We acknowledge that the version of the manuscript that was initially submitted had several spelling and grammatical errors. However, the authors have proof-read the revised manuscript and also sought an independent review from a colleague, Dr. Jerome Roy Semakula who recently excelled in his IELTS exam.

03 We note that you have provided funding information that is currently declared in your Funding Statement. However, funding information should not appear in the Acknowledgments section or other areas of your manuscript. We will only publish funding information present in the Funding Statement section of the online submission form. 

Please remove any funding-related text from the manuscript and let us know how you would like to update your Funding Statement. Thank you very much for raising this comment. We have deleted the funding information from the acknowledgment section in the revised manuscript. Thank you once again for the correction.

We request to maintain the funding information in the Funding Statement section of the online submission form

04 In your Data Availability statement, you have not specified where the minimal data set underlying the results described in your manuscript can be found. PLOS defines a study's minimal data set as the underlying data used to reach the conclusions drawn in the manuscript and any additional data required to replicate the reported study findings in their entirety. Thank you for raising this comment. We appreciate the spirit of sharing data for replication of the reported study findings and other benefits of data sharing. The data has been reported as quotes in the results section of the manuscript. This is the minimal data set used to reach the conclusions described in the manuscript.

05 We note that you have stated that you will provide repository information for your data at acceptance. Should your manuscript be accepted for publication, we will hold it until you provide the relevant accession numbers or DOIs necessary to access your data. If you wish to make changes to your Data Availability statement, please describe these changes in your cover letter and we will update your Data Availability statement to reflect the information you provide. Thank you very much for raising this concern. I have provided the minimal data set as quotes reported in the results section of the manuscript.

RESPONSES TO PEER REVIEW COMMENTS

ID COMMENT RESPONSES Page Line

01 Differentiate with responses. When I conduct focus groups, I usually have a design that differentiates people's roles to different focus groups. This way I can look at comparisons across groups with norms and concerns. You've presented the data as if all respondents - researchers, ethics committee members, and community members - all agree across every issue.

 Thank you very for raising this comment. This comment has helped us realize that we needed to clarify how the data were collected across the different study populations. We conducted focus group discussions with community representatives from five research institutions offering care and treatment to people living with HI. We also conducted key informant interviews with researchers involved in pharmacogenomic research and HIV treatment and in-depth interviews with members of research ethics committees who had prior experience in reviewing pharmacogenomic research for HIV treatment. We do agree with you that all the above categories of respondents have different roles and that their feedback has been presented with their roles. We have followed your recommendation on how to present the results and the discussion of our findings. The contributions from each category of the respondents have been well labelled at the end of each quote in the results section. 16 -29 NA

02 Identify what is important in the study and lean into it. You're providing stories on how various actors feel about concepts of community engagement. The report back on findings seems dry, and basically is reduced to "Community Engagement is important." Why and how are key questions that you do not approach directly. Thank you very much for raising this comment. We have followed your recommendation to focus on the research question when discussing our findings. We have provided detailed information to why and how community engagement plays an important role in promoting participants’ understanding of pharmacogenomic research results. 29-33 N/A

03 Avoid jargon. While you discuss pharmacogenomic research, it is often jargon. What makes this kind of research important for people living with HIV and who decides it's important? What are the ramifications of the research? There are times when you seem to conflate research with clinical practice (e.g. the individual wants their data from research). How is this possible? You make an assumption that all readers are up to date in this part of the field. Thank you very much for raising this comment. This comment has helped us realize that we needed to revise our discussion section. We have provided more detail about the importance of pharmacogenomic research to PLHIV and the researchers involved in the treatment of HIV/AIDS. We have also provided details on the ramifications of this research and tried to explain what pharmacogenomic research is to avoid jargon. 29-33 N/A

04 Barriers to Community Engagement are not identified. This is odd, as most researchers will discuss their frustrations openly (e.g. funding requirements, time, lack of expertise) yet there was little of that here (outside of language). Other barriers could be highlighted more. We have also provided information about other challenges that researchers experience when engaging communities in addition to some of them that were already mentioned. For example the absence of genetic counsellors who may be the most suitable professionals to determine the safe approaches and methods of communicating genomic and pharmacogenomic research information in different communities. In addition, we have added information about the limited funding for community engagement activities in some research institutions. 32 624-625

05 Please remove any funding-related text from the manuscript and let us know how you would like to update your Funding Statement Thank you very much for your feedback. We have revised the acknowledgment section as advised. 34 670-672

---

## [Editor Report · Decision Letter 1]

13 Dec 2023

PONE-D-23-22714R1The role of community engagement in promoting research participants’ understanding of pharmacogenomic research results: Perspectives of stakeholders involved in HIV/AIDS research and treatmentPLOS ONE

Dear Dr. Nabukenya,

Thank you for submitting your manuscript to PLOS ONE. After careful consideration, we feel that it has merit but does not fully meet PLOS ONE’s publication criteria as it currently stands. Therefore, we invite you to submit a revised version of the manuscript that addresses the points raised during the review process.

We look forward to receiving your revised manuscript.

Kind regards,

Tahir Turk, PhD

Academic Editor

PLOS ONE

Additional Editor Comments:

Dear Dr Nabukenya

Thank-you for addressing the reviewer feedback on your manuscript and making amendments on the funding statement and acknowledgements. However, when submitting a manuscript, authors must provide a Data Availability Statement describing compliance with PLOS' data policy. If the article is accepted for publication, the Data Availability Statement will be published as part of the article. PLOS believes that sharing data fosters scientific progress. Data availability allows and facilitates:

- Validation, replication, reanalysis, new analysis, reinterpretation or inclusion into meta-analyses;

- Reproducibility of research;

- Efforts to ensure data are archived, increasing the value of the investment made in funding scientific research;

- Reduction of the burden on authors in preserving and finding old data, and managing data access requests;

- Citation and linking of research data and their associated articles, enhancing visibility and ensuring recognition for authors, data producers and curators.

Acceptable data sharing methods are listed in PLSO ONE's Data Availability information section https://journals.plos.org/plosone/s/data-availability, which provides guidance for authors as to what must be included in their Data Availability Statement and how to follow best practices in research reporting. Publication is conditional on compliance with this policy. Therefore, PLOS strongly recommends sharing data in a repository whenever possible. Data repositories improve discoverability and accessibility, ensure long-term preservation, and lead to increased attention for the research. The Minimal Data Set Definition consist of the data required to replicate all study findings reported in the article, as well as related metadata and methods. Additionally, PLOS requires that authors comply with field-specific standards for preparation, recording, and deposition of data when applicable.

Given these requirements we believe the dFGD data-set and the KIIs and IDI datasets from your study should be provided in a public data repository with relevant DOI identifiers. Failing compliance with the minimal dat set requirements, will mean your manuscript cannot be accepted for publishing by our Journal.

---

## [Author Response · Author response to Decision Letter 1]

4 Jan 2024

2 January 2024

Academic Editor,

PLOS ONE Journal

Dear Dr Tahir Turk,

RE: Response to reviewers’ comments.

On behalf of the authors, I take this opportunity to appreciate you and the reviewers for sparing time to review and provide insightful feedback to improve the quality of our manuscript entitled “The role of community engagement in promoting research participants’ understanding of pharmacogenomic research results: Perspectives of stakeholders involved in HIV/AIDS research and treatment” PONE-D-23-22714R1 for consideration by the PLOS ONE journal.

I hereby submit a point-by-point response to the peer review comments of the above named manuscript (attached).

We have positively responded to all the concerns raised by the reviewers and we believe that their feedback has greatly improved the quality and readability of our manuscript.

Yours sincerely,

Sylvia Nabukenya

Email: nabukenyas89@gmail.com

RESPONSE TO ADDITIONAL JOURNAL REQUIREMENTS

ID COMMENTS RESPONSES

01 Please ensure that your manuscript meets PLOS ONE's style requirements, including those for file naming. The PLOS ONE style templates can be found at 

Thank you very much for raising this comment and providing the links to the style templates. We have revised the format of the manuscript as guided.

02 We suggest you thoroughly copyedit your manuscript for language usage, spelling, and grammar. If you do not know anyone who can help you do this, you may wish to consider employing a professional scientific editing service.

 The name of the colleague or the details of the professional service that edited your manuscript Thank you very much for raising this comment. We acknowledge that the version of the manuscript that was initially submitted had several spelling and grammatical errors. However, the authors have proof-read the revised manuscript and also sought an independent review from a colleague, Dr. Jerome Roy Semakula who recently excelled in his IELTS exam.

03 We note that you have provided funding information that is currently declared in your Funding Statement. However, funding information should not appear in the Acknowledgments section or other areas of your manuscript. We will only publish funding information present in the Funding Statement section of the online submission form. 

Please remove any funding-related text from the manuscript and let us know how you would like to update your Funding Statement. Thank you very much for raising this comment. We have deleted the funding information from the acknowledgment section in the revised manuscript. Thank you once again for the correction.

We request to maintain the funding information in the Funding Statement section of the online submission form

04 In your Data Availability statement, you have not specified where the minimal data set underlying the results described in your manuscript can be found. PLOS defines a study's minimal data set as the underlying data used to reach the conclusions drawn in the manuscript and any additional data required to replicate the reported study findings in their entirety. Thank you for raising this comment. We appreciate the spirit of sharing data for replication of the reported study findings and other benefits of data sharing. The minimal data sets used to reach the conclusions described in the manuscript will be provided once accepted for publication.

05 We note that you have stated that you will provide repository information for your data at acceptance. Should your manuscript be accepted for publication, we will hold it until you provide the relevant accession numbers or DOIs necessary to access your data. If you wish to make changes to your Data Availability statement, please describe these changes in your cover letter and we will update your Data Availability statement to reflect the information you provide. Thank you very much for raising this comment. Once the manuscript is accepted for publication, we shall provide the dFGD, KIIs and IDI data sets.

RESPONSES TO PEER REVIEW COMMENTS

ID COMMENT RESPONSES Page Line

01 Differentiate with responses. When I conduct focus groups, I usually have a design that differentiates people's roles to different focus groups. This way I can look at comparisons across groups with norms and concerns. You've presented the data as if all respondents - researchers, ethics committee members, and community members - all agree across every issue.

 Thank you very for raising this comment. This comment has helped us realize that we needed to clarify how the data were collected across the different study populations. We conducted focus group discussions with community representatives from five research institutions offering care and treatment to people living with HI. We also conducted key informant interviews with researchers involved in pharmacogenomic research and HIV treatment and in-depth interviews with members of research ethics committees who had prior experience in reviewing pharmacogenomic research for HIV treatment. We do agree with you that all the above categories of respondents have different roles and that their feedback has been presented with their roles. We have followed your recommendation on how to present the results and the discussion of our findings. The contributions from each category of the respondents have been well labelled at the end of each quote in the results section. 16 -29 NA

02 Identify what is important in the study and lean into it. You're providing stories on how various actors feel about concepts of community engagement. The report back on findings seems dry, and basically is reduced to "Community Engagement is important." Why and how are key questions that you do not approach directly. Thank you very much for raising this comment. We have followed your recommendation to focus on the research question when discussing our findings. We have provided detailed information to why and how community engagement plays an important role in promoting participants’ understanding of pharmacogenomic research results. 29-33 N/A

03 Avoid jargon. While you discuss pharmacogenomic research, it is often jargon. What makes this kind of research important for people living with HIV and who decides it's important? What are the ramifications of the research? There are times when you seem to conflate research with clinical practice (e.g. the individual wants their data from research). How is this possible? You make an assumption that all readers are up to date in this part of the field. Thank you very much for raising this comment. This comment has helped us realize that we needed to revise our discussion section. We have provided more detail about the importance of pharmacogenomic research to PLHIV and the researchers involved in the treatment of HIV/AIDS. We have also provided details on the ramifications of this research and tried to explain what pharmacogenomic research is to avoid jargon. 29-33 N/A

04 Barriers to Community Engagement are not identified. This is odd, as most researchers will discuss their frustrations openly (e.g. funding requirements, time, lack of expertise) yet there was little of that here (outside of language). Other barriers could be highlighted more. We have also provided information about other challenges that researchers experience when engaging communities in addition to some of them that were already mentioned. For example the absence of genetic counsellors who may be the most suitable professionals to determine the safe approaches and methods of communicating genomic and pharmacogenomic research information in different communities. In addition, we have added information about the limited funding for community engagement activities in some research institutions. 32 624-625

05 Please remove any funding-related text from the manuscript and let us know how you would like to update your Funding Statement Thank you very much for your feedback. We have revised the acknowledgment section as advised. 34 670-672

---

## [Decision Letter · Decision Letter 2]

6 Feb 2024

The role of community engagement in promoting research participants’ understanding of pharmacogenomic research results: Perspectives of stakeholders involved in HIV/AIDS research and treatment

PONE-D-23-22714R2

Dear Dr.Nabukenya,

We’re pleased to inform you that your manuscript has been judged scientifically suitable for publication and will be formally accepted for publication once it meets all outstanding technical requirements.

Kind regards,

Tahir Turk, PhD

Academic Editor

PLOS ONE

**Additional Editor Comments **

Thank-you for addressing the concerns of the reviewer in the revised manuscript. Although we note that the authors have not included a COREQ or similar qualitative research checklist with the manuscript we note that a number of the checklist requirements have been addressed through the revision. We are also pleased to see practical and pragmatic recommendations emanating from the study.

**Reviewers' **Comments to the Author****

Reviewer #1: Thank you for addressing my concerns in my review. I found the current version of the study to be interesting and compelling. All comments have been addressed

---

## [Editor Report · Acceptance letter]

22 Mar 2024

PONE-D-23-22714R2 

PLOS ONE

Dear Dr. Nabukenya, 

I'm pleased to inform you that your manuscript has been deemed suitable for publication in PLOS ONE. Congratulations! Your manuscript is now being handed over to our production team.

Kind regards, 

on behalf of

Dr. Tahir Turk 

Academic Editor

PLOS ONE